# Fault-Tolerant Control for Carrier-Based UAV Based on Sliding Mode Method

Zhuoer Yao [1], Zi Kan [1,*] , Chong Zhen [2], Haoyuan Shao [1] and Daochun Li [1,*]

[1] School of Aeronautic Science and Engineering, Beihang University, Beijing 100191, China
[2] AVIC Shenyang Aircraft Design and Research Institute, Shenyang 110087, China
* Correspondence: kanzi2017@buaa.edu.cn (Z.K.); lidc@buaa.edu.cn (D.L.)

**Abstract:** To enable a carrier-based unmanned aerial vehicle (UAV) to track the desired glide trajectory and safely land on the deck with the presence of system faults, this paper proposes a neural network-based adaptive sliding mode fault-tolerant control (NASFTC) method. Firstly, the dynamic model of the carrier-based UAV, the actuator fault model, the additional unknown fault model, and the control framework of the automatic carrier landing system (ACLS) were developed. Subsequently, controllers for both longitudinal and lateral channels were designed by using the NASFTC method. The controller consists of three parts: the adaptive laws for compensating the actuator faults, the RBF neural network for compensating the additional unknown faults, and the sliding mode method for ensuring overall trajectory tracking. Then, the Lyapunov function theorem was applied to carry out the stability analysis. Finally, comparative simulations under three different scenarios were conducted. The comparative results show the effectiveness of the proposed NASFTC method, which has fault-tolerant ability and can successfully control the aircraft to execute carrier landing task regardless of the actuator partial loss fault and the additional unknown fault.

**Keywords:** carrier-based unmanned aerial vehicle; system fault; sliding mode; fault-tolerant control





## 1. Introduction

Carrier-based unmanned aerial vehicles (UAVs) have attracted wide attention because they can take the place of humans to perform dangerous tasks, and the presence of an aircraft carrier has greatly improved their combat range. However, the carrier landing task is known as "dancing on the knife point" due to its severe difficulty. To land the aircraft safely on the deck, various studies have been investigated [1–4]. The carrier landing issue can be essentially considered as a three-dimensional moving path following problem, where the aircraft is required to follow a trajectory attached to a moving carrier [5]. To solve this moving path following problem, some modern control methods were employed to design an automatic carrier landing system (ACLS), such as adaptive constrained backstepping control [6], direct lift control [7], sliding mode control [8], and adaptive super-twisting control [9]. Furthermore, there are external disturbances such as air wake and deck motion, which can cause an undesirable effect on the carrier landing. A disturbance observer was developed to estimate the air-wake disturbance and ensure a precise landing [10]. The particle filtering method has also been used to predict deck motion and correct the reference glide trajectory [11].

The above research assumes that the aircraft is in good condition without faults. However, carrier-based UAV is a complex synthesis with numerous mechanisms. When the aircraft executes a landing task in hostile environments, such as strong wind and waves, it is inevitable to have faults such as actuator damage. Once the faults occur, it significantly increases the risk of landing failure. Generally, the faults of the aircraft can be divided into three categories: sensor fault, actuator fault, and structural fault. The reasons for these faults can be equipment damage and environmental changes [12]. Without appropriate control methods, each of these faults can have catastrophic consequences.

Therefore, fault-tolerant control (FTC) of carrier-based UAV is necessary and essential. In recent years, abundant investigations into FTC have been conducted [13–16]. A robust FTC was proposed in Reference [17], and numerical simulations showed that this method was effective for flexible spacecraft with actuator partial loss. By increasing the control signals to actuators and reducing cost function, the passive FTC method that was designed in Reference [18] can successfully keep an electric vehicle stable with three types of actuator faults. Another passive fault-tolerant flight controller is presented for actuator failures [19], in which the peak–peak gain concept is adopted. The results demonstrated this controller is effective despite the presence of actuator failures.

However, these are all passive FTC methods. The main characteristic of passive FTC is by improving the robustness of the controller, and the system will be insensitive to faults. The advantage of this method is that there is no need for fault detection, and the same controller can be used for both normal and faulty states. However, the adaptability of the passive FTC method for faults is very limited. Once the faults break the tolerance boundary, it will cause control failure, which is usually unacceptable.

To improve the system's adaptability to different faults, the active FTC method has attracted attention [20–23]. A learning observer with adaptive ability is proposed in Reference [24], which can actively detect and reconstruct actuator faults where the space-craft attitude is stabilized by the estimation of this learning observer and backstepping control method. Another study considered the thruster system faults of an underwater vehicle and developed a finite-time observer to make a real-time estimation of the thruster failures. After the estimation, the active compensation of the thruster system is made by control allocation [25]. Reference [26] proposed an adaptive control mechanism for a ducted-fan robot to estimate the error that is caused by actuator failure while the designed robust loop realized the overall stability. These active FTC methods can automatically adjust the controller parameters and even change the controller structure according to the fault condition, which equips the controller with a more remarkable fault tolerance ability. Therefore, this paper developed an active FTC method to solve the faulty carrier landing problem for the carrier-based UAV. This method can effectively deal with faults by actively adjusting control parameters and making compensations.

In the field of carrier-based aircraft landing control, the application of FTC is still in its infancy with few investigations. A fault-tolerant method was added to the longitudinal controller to enhance the trajectory tracking ability, where a neural network is used to compensate for actuator faults [27]. An adaptive FTC method was developed in Reference [28], which utilized an adaptive law to compensate for the actuator faults. Simulation results demonstrate its effectiveness for both parameterized and unparameterized faults. Another FTC method for actuator faults was proposed in Reference [29], where the faults are mod-eled as both partial loss and lock-in-place. Although there are some pioneering studies on carrier-based aircraft faulty landing problems, many of them only focus on one fault. While during the carrier landing process, there can be a combination of multiple faults. In this paper, both the actuator fault and the additional unknown fault are considered, which puts forward a higher requirement for the controller. The actuator faults will make the control surfaces fail to respond to the instructions, and the additional unknown faults will inject faulty information into the UAV. All these faults can lead to the failure of the carrier landing. To achieve a safe carrier landing with multiple faults, the following contributions are made in this paper.

(1) A neural network-based adaptive sliding mode fault-tolerant control (NASFTC) method is proposed. The adaptive laws are used to automatically adjust the controller parameters according to the actuator faults. The neural network is used to predict and compensate for the error that is induced by additional unknown faults and the function of sliding mode is to ensure the overall tracking of the desired gliding trajectory. By the above characteristics, the proposed control method can achieve a safe carrier landing with the presence of multiple faults.

(2) An automatic carrier landing system (ACLS) control framework based on the proposed NASFTC method for the carrier-based UAV is established. This framework consists of a glide path generation subsystem, a guidance subsystem, a flight controller subsystem and a faulty aircraft subsystem where both longitudinal and lateral channels are taken into consideration.

The rest of this paper is arranged as follows. Section 2 presents the dynamic model of carrier-based UAV, the actuator fault model, the additional unknown fault model and the ACLS control framework. The design details of the controller are given in Section 3, which includes the design of the neural network, longitudinal controller, lateral controller, and approach power compensation system. The simulation experiments and results discussions are carried out in Section 4, where a total of three scenarios are considered. Finally, the conclusions are presented in Section 5.

## 2. Problem Formulation

In this section, the UAV model is established, the fault models are introduced to the UAV for controller design, and the ACLS framework is presented to show how to command an aircraft to land on the carrier.

### 2.1. UAV Model

The carrier-based UAV that is considered in this paper is equipped with an elevator, aileron, and rudder. The actuators are hydraulic, and a 5% control gain reduction is considered. During the landing process, the desired sideslip angle is set at zero. Therefore, the dynamics model of the carrier-based aircraft can be written as follows [30]:

$$\begin{cases} \dot{x}_g = V \cos\gamma\cos\chi \\ \dot{y}_g = V \cos\gamma\sin\chi \\ \dot{z}_g = -V\sin\gamma \end{cases} \tag{1}$$

$$\begin{cases} \dot{u} = rv - qw - g\sin\theta + (F_X + T_X)/m \\ \dot{v} = -ru + pw - g\sin\phi\cos\theta + (F_Y + T_Y)/m \\ \dot{w} = qu - pv + g\cos\phi\cos\theta + (F_Z + T_Z)/m \end{cases} \tag{2}$$

$$\begin{cases} \dot{\phi} = p + \tan\theta(q\sin\phi + r\cos\phi) \\ \dot{\theta} = q\cos\phi - r\sin\phi \\ \dot{\beta} = p\sin\alpha - r\cos\alpha + \frac{1}{mv}(F_Y + T_Y) \end{cases} \tag{3}$$

$$\begin{cases} \dot{p} = I_1 pq + I_2 qr + I_3 L + I_4 N \\ \dot{q} = I_5 pr - I_6(p^2 - r^2) + I_7 M \\ \dot{r} = I_8 pq - I_1 qr + I_4 L + I_9 N \end{cases} \tag{4}$$

where $x_g, y_g, z_g$ denotes the positions of UAV, $V, \gamma, \chi$ denotes the ground velocity, heading angle and climbing angle, respectively, $u, v, w$ denotes the velocity in the body-fixed reference frame, $p, q, r$ denotes the angular rates in the body-fixed reference frame, $\phi, \theta, \alpha, \beta$ denotes the roll angle, pitch angle, angle of attack, and sideslip angle, respectively, $F_X, F_Y, F_Z$ denotes the aerodynamic forces along the axis in the body-fixed reference frame, $T_X, T_Y, T_Z$ denotes the thrust along the axis in the body-fixed reference frame, and $L, M, N$ denotes the moments along the axis in the body-fixed reference frame.

Assuming $\mathbf{x_1} = \begin{bmatrix} \phi & \theta & \beta \end{bmatrix}^T$ and $\mathbf{x_2} = \begin{bmatrix} p & q & r \end{bmatrix}^T$, Equations (3) and (4) can be rewritten into affine form as follows:

$$\begin{cases} \dot{\mathbf{x}}_1(t) = \mathbf{A} + \mathbf{B}\mathbf{x}_2(t) \\ \dot{\mathbf{x}}_2(t) = \mathbf{F} + \mathbf{G}\mathbf{u}(t) \end{cases} \tag{5}$$

where $\mathbf{u} = \begin{bmatrix} \delta_a & \delta_e & \delta_r \end{bmatrix}^T$ denotes deflections of the aileron, elevator, and rudder, respectively; and $\mathbf{A}$, $\mathbf{B}$, $\mathbf{F}$ and $\mathbf{G}$ are system matrixes of the UAV.

### 2.2. Actuator Partial Loss Model and Additional Unknown Fault Model

In this paper, two different kinds of faults are considered, which are the actuator fault and the additional unknown fault. During the carrier landing process, actuator faults may occur due to various factors, which will lead to landing failure. Actuator partial loss is one of the common faults of actuators. This fault represents a reduction of control gain, resulting in a deviation of the command signal, thereby weakening the actuator function. Generally, the actuator partial loss fault can be expressed by:

$$\mathbf{u_f(t)} = \mathbf{\Sigma(t)u(t)}, t \geq t_f \tag{6}$$

where $t_f$ denotes the failure time, which is unknown to the system, and $\mathbf{\Sigma(t)}$ is the actuator effectiveness matrix. For the carrier-based UAV that is discussed in this paper, there are three actuators: aileron, rudder, and elevator. Therefore, the actuator effectiveness matrix is $\mathbf{\Sigma(t)} = diag(\sigma_1(t), \sigma_2(t), \sigma_3(t))$ with $0 < \sigma_i(t) \leq 1$, where $\sigma_i(t) = 1$ means the corresponding actuator works normally and $0 < \sigma_i(t) < 1$ means that partial loss has occurred on the corresponding actuator.

Apart from the actuator fault, an additional unknown fault is also considered in this paper. The expression of this additional unknown fault is regarded as a nonlinear function, which is:

$$\mathbf{d(t, x_1, x_2)} = \begin{bmatrix} d_1 & d_2 & d_3 \end{bmatrix}^T \tag{7}$$

where $d_i < D, i = 1, 2, 3$ with $D$ being a known constant, indicating the additional unknown fault has a certain boundary.

By introducing the models of actuator partial loss and additional unknown fault into Equation (5), the flight state-space equation with faults for controller design is established as:

$$\begin{cases} \mathbf{\dot{x}_1(t)} = \mathbf{A} + \mathbf{Bx_2(t)} + \mathbf{d(t, x_1, x_2)} \\ \mathbf{\dot{x}_2(t)} = \mathbf{F} + \mathbf{\Sigma Gu(t)} \end{cases} \tag{8}$$

### 2.3. ACLS Control Framework

The ACLS control framework is composed of a glide path generation subsystem, a guidance subsystem, a flight controller subsystem and a faulty aircraft subsystem, as depicted in Figure 1.

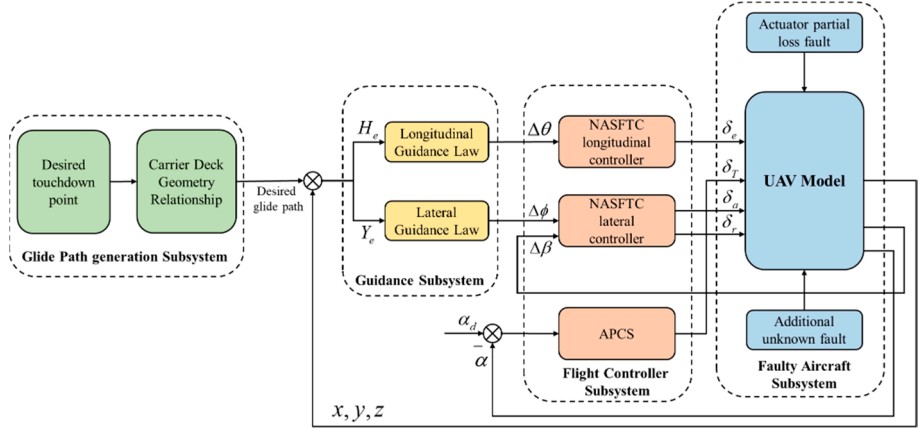

**Figure 1.** The control framework of ACLS.

The function of the glide path generation subsystem is to output the longitudinal and lateral tracking errors between the UAV and the desired glide path and localizer path in lateral meaning. Since the desired glide path will be directly affected by the movement of

the carrier, it is important to establish their geometric relationship. The desired touchdown point is defined as $\left( x_{dtp}, y_{dtp}, z_{dtp} \right)$ and the aircraft's center of mass is defined as $(x_b, y_b, z_b)$. The relative position of the aircraft and the carrier is given by:

$$\begin{bmatrix} x_r \\ y_r \\ z_r \end{bmatrix} = \mathbf{L_{dg}}(\phi_c, \theta_c + \chi_c, \psi_c + \psi_{fd}) \begin{bmatrix} x_b - x_{dtp} \\ y_b - y_{dtp} \\ z_b - z_{dtp} \end{bmatrix} \tag{9}$$

where $\mathbf{L_{dg}}$ is the transfer matrix from earth-fixed inertial reference frame to carrier deck reference frame; $\phi_c, \theta_c, \chi_c, \psi_c$ and $\psi_{fd}$ are the carrier's rolling angle, pitching angle, heading angle, yawing angle, and flight deck angle, respectively. Therefore, as shown in Figure 2, for the given desired glide slope angle $\gamma_g$, the tracking error can be expressed by:

$$\begin{bmatrix} H_e \\ Y_e \end{bmatrix} = \begin{bmatrix} z_r - x_r \tan \sigma_p \\ y_r \end{bmatrix} \tag{10}$$

where $\sigma_p$ is the angle between the desired path and the carrier, which is given by:

$$\tan \sigma_p = \frac{\sin \gamma_p}{\cos \gamma_p - V_c / V_p} \tag{11}$$

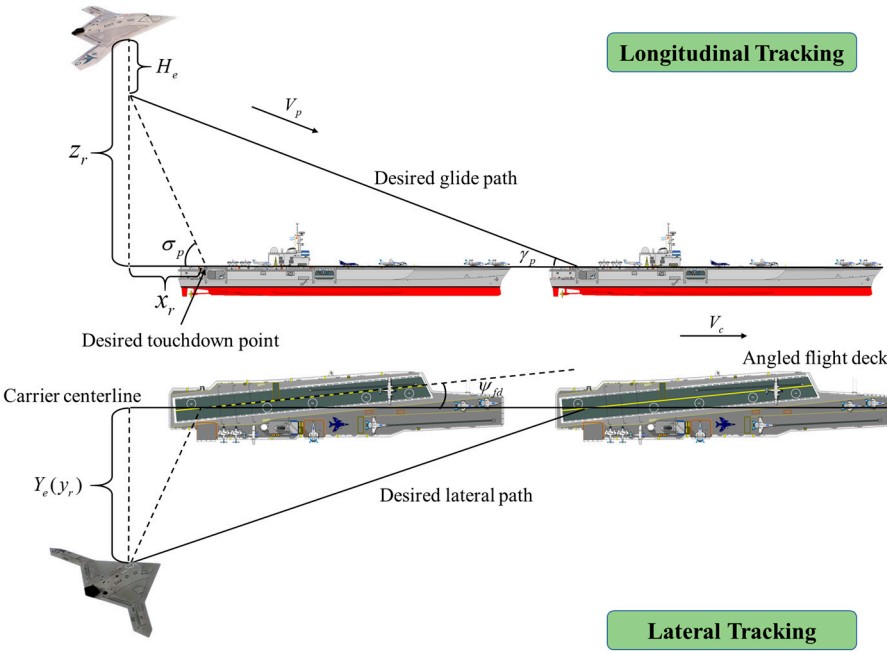

**Figure 2.** Schematic diagram of the landing process.

The guidance subsystem is used to transform the displacement deviations into the angle control commands. Generally, the guidance subsystem can be divided into the longitudinal channel and the lateral channel. In the longitudinal channel, the Hdot guidance law is adopted, which can generate the pitch angle command $\theta_c$ for height tracking [31]. The key point of the Hdot guidance law is to keep the angle of attack (AoA) $\alpha$ unchanged. In the lateral guidance channel, the PID method is adopted, which can generate the roll angle command $\phi_c$ and yaw angle command $\psi_c$. The key point of the lateral channel is to keep the sideslip angle $\beta$ at zero.

The faulty aircraft subsystem includes a 6-DOF aircraft model, actuator partial loss fault model and an additional unknown fault model. The occurrence time of the faults are unknown to the system, the same as the values of these two faults. After introducing the faults into the aircraft, accurate operations of landing cannot be guaranteed, increasing the

risk of landing failure. Therefore, it is essential to empower the controller to compensate for the uncertainties that are caused by the faults.

The flight controller subsystem consists of a longitudinal flight controller, a lateral flight controller, and an approach power compensation system (APCS). The aim of APCS is to output a proper $\delta_T$ to maintain the AoA, so that the Hdot guidance law can operate normally. Meanwhile, the longitudinal and lateral flight controllers output the control commands $\delta_e$, $\delta_a$, and $\delta_r$. To eliminate the influence of the faults, the flight controllers are designed by the NASFTC method. The functions of the adaptive law and the RBF neural network are to compensate the actuator partial loss and additional unknown fault, respectively. Simulation results demonstrate that the controllers that are proposed in this paper can effectively accommodate multiple faults and ensure the aircraft tracks the desired glide path.

## 3. Controller Design

In this section, the detailed design steps of the RBF neural network, the flight controllers, and the approach power compensation system are provided.

### 3.1. RBF Neural Network

In this paper, the RBF neural network is used to compensate the additional unknown faults. Many studies have proven that neural networks have excellent learning and approximation ability [32–35]. Among all these neural networks, the RBF neural network is superior to the traditional neural networks, mainly in the characteristics of robustness and tolerance [36]. The structure of the RBF neural network is shown in Figure 3. Assuming the approaching error is $\varepsilon = [\varepsilon_1, \varepsilon_2, \ldots, \varepsilon_m]^T$, the nonlinear additional unknown fault $\mathbf{d}(\mathbf{t}, \mathbf{x_1}, \mathbf{x_2})$ can be written as:

$$\mathbf{d}(\mathbf{t}, \mathbf{x_1}, \mathbf{x_2}) = \mathbf{W}^{*\mathbf{T}}\mathbf{h}(\mathbf{t}, \mathbf{x_1}, \mathbf{x_2}) + \varepsilon \tag{12}$$

$$h_i(t, x_1, x_2) = \exp(-\frac{\|x - c_i\|^2}{2b_i^2}), i = 1, 2, \ldots, m \tag{13}$$

where $\mathbf{h}(\mathbf{t}, \mathbf{x_1}, \mathbf{x_2}) = [h_i]^T$ is the Gaussian potential function output of the neural network, $\mathbf{W}^{*T} = [W_1^*, W_2^*, \ldots, W_m^*]^T$ is the ideal weight, and $c_i$ and $b_i$ are the center and spread of the gaussian potential function, respectively.

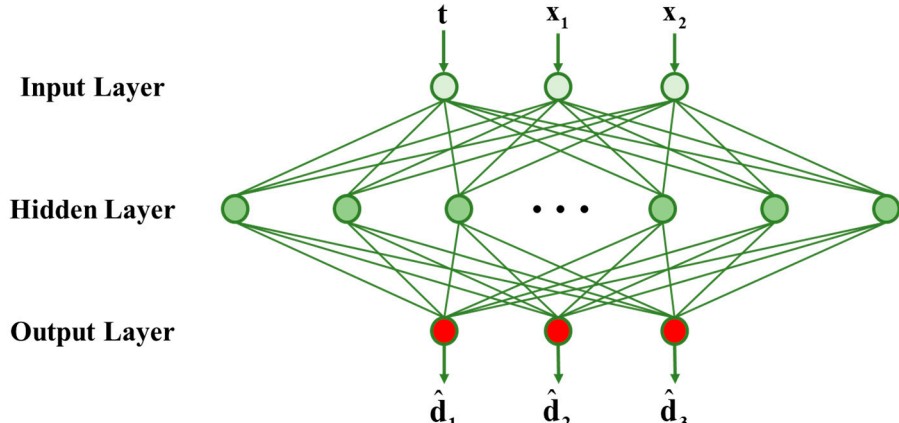

**Figure 3.** Structure of RBF neural network.

Therefore, the additional unknown fault can be estimated by:

$$\hat{\mathbf{d}}(\mathbf{t}, \mathbf{x_1}, \mathbf{x_2}) = \hat{\mathbf{W}}^{\mathbf{T}}\mathbf{h}(\mathbf{t}, \mathbf{x_1}, \mathbf{x_2}) \tag{14}$$

where $\hat{\mathbf{W}}^{\mathbf{T}}$ is the estimation weight.

The ideal weight $\mathbf{W}^{*T}$ can be calculated by:

$$\mathbf{W}^{*T} = \left[ \arg \min_{\hat{W}^T \in \Omega_d} \left( \sup_{x \in \Omega_x} \left| \hat{\mathbf{d}}(\mathbf{t}, \mathbf{x_1}, \mathbf{x_2}) - \mathbf{d}(\mathbf{t}, \mathbf{x_1}, \mathbf{x_2}) \right| \right) \right]^T \tag{15}$$

### 3.2. Controller Design

In this section, the detailed design steps of NASFTC are provided. As mentioned in Section 2, with the help of the glide path generation subsystem and guidance subsystem, the obtained trajectory tracking errors can be transformed into the attitude control commands. Therefore, the landing control problem is transformed into an attitude control problem. The structure of the controller is shown in Figure 4.

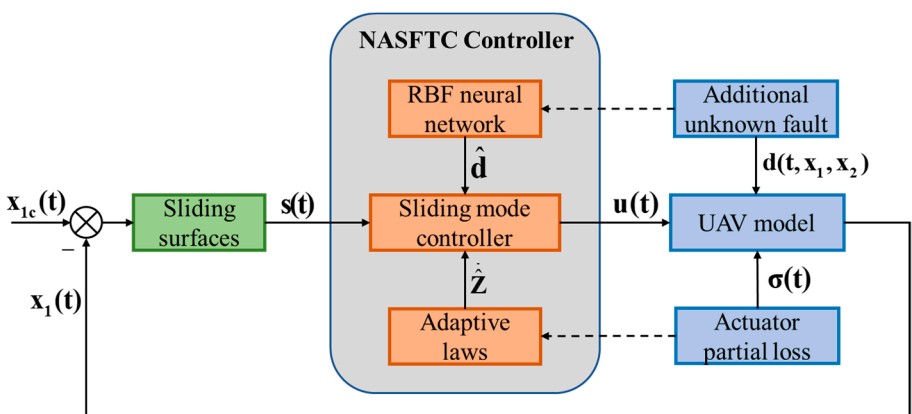

**Figure 4.** Structure of NASFTC.

Normally, the dynamics of the angular variables $\mathbf{x_1} = \begin{bmatrix} \phi & \theta & \beta \end{bmatrix}^T$ are slower than the angular velocity variables $\mathbf{x_2} = \begin{bmatrix} p & q & r \end{bmatrix}^T$, which allows us to employ the timescale separation principle. Equation (8) shows that the faulty flight dynamic model is a cascaded system structure with two equations, which means the command input of the first equation can be the reference signal of the second equation.

For the first equation in Equation (8) $\dot{\mathbf{x}}_1(\mathbf{t}) = \mathbf{A} + \mathbf{B}\mathbf{x_2}(\mathbf{t}) + \mathbf{d}(\mathbf{t}, \mathbf{x_1}, \mathbf{x_2})$, assuming the control commands that are obtained by the guidance subsystem are $\begin{bmatrix} \phi_d & \theta_d \end{bmatrix}$. The desired sideslip angle is set at zero. Therefore, the desired attitude control commands are $\mathbf{x_{1d}}(\mathbf{t}) = \begin{bmatrix} \phi_d & \theta_d & \beta_d \end{bmatrix}$. The tracking error is defined as follows:

$$\mathbf{e_1}(\mathbf{t}) = \mathbf{x_1}(\mathbf{t}) - \mathbf{x_{1d}}(\mathbf{t}) \tag{16}$$

The sliding mode method is used to design the controller [37]. Firstly, the sliding surfaces are defined in the form of:

$$\mathbf{s_1}(\mathbf{t}) = \mathbf{e_1}(\mathbf{t}) + \mathbf{K_1} \int_0^t \mathbf{e_1}(\mathbf{t})\mathbf{d}\tau \tag{17}$$

Taking the derivative of $\mathbf{S_1}(\mathbf{t})$ with respect to time yields:

$$\dot{\mathbf{s}}_1 = \dot{\mathbf{e}}_1 + \mathbf{K_1}\mathbf{e_1} = \dot{\mathbf{x}}_1 - \dot{\mathbf{x}}_{1d} + \mathbf{K_1}\mathbf{e_1} \tag{18}$$

Substituting Equation (8) into Equation (18) yields:

$$\dot{\mathbf{s}}_1 = \mathbf{A} + \mathbf{B}\mathbf{x_2} + \mathbf{d} - \dot{\mathbf{x}}_{1d} + \mathbf{K_1}\mathbf{e_1} \tag{19}$$

Using the RBF neural network to approximate the additional unknown faults $\mathbf{d(t, x_1, x_2)}$, the approximation can be written as:

$$\hat{\mathbf{d}}(\mathbf{t, x_1, x_2}) = \hat{\mathbf{W}}^{\mathbf{T}}\mathbf{h(t, x_1, x_2)} \tag{20}$$

The fault approximation error is defined as:

$$\begin{aligned}
\widetilde{\mathbf{d}}(\mathbf{t, x_1, x_2}) &= \mathbf{d(t, x_1, x_2)} - \hat{\mathbf{d}}(\mathbf{t, x_1, x_2}) \\
&= \mathbf{W^{*T}h(t, x_1, x_2)} + \varepsilon - \hat{\mathbf{W}}^{\mathbf{T}}\mathbf{h(t, x_1, x_2)} \\
&= \widetilde{\mathbf{W}}^{\mathbf{T}}\mathbf{h(t, x_1, x_2)}
\end{aligned} \tag{21}$$

where $\boldsymbol{\varepsilon} = \begin{bmatrix} \varepsilon_1 & \varepsilon_2 & \varepsilon_3 \end{bmatrix}^T$ is the neural network error.

Taking the command input $\mathbf{x_2}$ in the form of:

$$\mathbf{x_2} = \mathbf{B}^{-1}(-\mathbf{A} - \hat{\mathbf{d}} + \dot{\mathbf{x}}_{\mathbf{1d}} - \mathbf{K_1 e_1} - \mathbf{C_1 s_1} - \mathbf{E_1 sgn(s_1)}) \tag{22}$$

where $\mathbf{E_1} = diag(\eta_{11}, \eta_{21}, \eta_{31})$ is constant matrixes with $\eta_{i1} > \varepsilon_i$.

**Theorem 1.** *For the first equation in Equation (8), supposing* $\dot{\hat{\mathbf{W}}}^{\mathbf{T}} = \mu \mathbf{s_1}(\mathbf{t})\mathbf{h(t, x_1, x_2)}$, *where $\mu$ is a constant coefficient. If the command input is chosen in the form of Equation (18), the tracking error* $\mathbf{e_1(t)}$ *can asymptotically converge to zero.*

**Proof 1.** Selecting a Lyapunov candidate function as follows:

$$V_1 = \frac{1}{2}\mathbf{s_1}^{\mathbf{T}}\mathbf{s_1} + \frac{1}{2\mu}\widetilde{\mathbf{W}}^T\widetilde{\mathbf{W}} \tag{23}$$

.□

The derivative of $V_1$ is:

$$\dot{V}_1 = \mathbf{s_1}^{\mathbf{T}}\dot{\mathbf{s}}_{\mathbf{1}} + \frac{1}{\mu}\widetilde{\mathbf{W}}^T\dot{\widetilde{\mathbf{W}}} \tag{24}$$

Substituting Equations (19), (21) and (22) into Equation (24) yields:

$$\begin{aligned}
\dot{V}_1 &= \mathbf{s_1}^{\mathbf{T}}\dot{\mathbf{s}}_{\mathbf{1}} + \frac{1}{\mu}\widetilde{\mathbf{W}}^T\dot{\widetilde{\mathbf{W}}} = \mathbf{s_1}^{\mathbf{T}}\dot{\mathbf{s}}_{\mathbf{1}} - \frac{1}{\mu}\widetilde{\mathbf{W}}^T\dot{\hat{\mathbf{W}}} \\
&= \mathbf{s_1}^{\mathbf{T}}(\mathbf{A} + \mathbf{B}\mathbf{x_2} + \mathbf{d} - \dot{\mathbf{x}}_{\mathbf{1d}} + \mathbf{K_1 e_1}) - \frac{1}{\mu}\widetilde{\mathbf{W}}^T\dot{\hat{\mathbf{W}}} \\
&= \mathbf{s_1}^{\mathbf{T}}(\widetilde{\mathbf{d}} - \mathbf{C_1 s_1} - \mathbf{E_1 sgn(s_1)}) - \frac{1}{\mu}\widetilde{\mathbf{W}}^T\dot{\hat{\mathbf{W}}} \\
&= \mathbf{s_1}^{\mathbf{T}}(\varepsilon - \mathbf{C_1 s_1} - \mathbf{E_1 sgn(s_1)}) + \widetilde{\mathbf{W}}^T(\mathbf{s_1 h(x)} - \frac{1}{\mu}\dot{\hat{\mathbf{W}}})
\end{aligned} \tag{25}$$

Substituting the adaptive law $\dot{\hat{\mathbf{W}}}^{\mathbf{T}} = \mu \mathbf{s_1}(\mathbf{t})\mathbf{h(t, x_1, x_2)}$ into Equation (25) yields:

$$\begin{aligned}
\dot{V}_1 &= \mathbf{s_1}^{\mathbf{T}}(\varepsilon - \mathbf{C_1 s_1} - \mathbf{E_1 sgn(s_1)}) \\
&= -\mathbf{C_1 s_1}^2 + \mathbf{s_1}^{\mathbf{T}}\varepsilon - |\mathbf{s_1}^{\mathbf{T}}|\mathbf{E_1} \leq -\mathbf{C_1 s_1}^2 \leq 0
\end{aligned} \tag{26}$$

According to the Lyapunov and LaSalle-Yoshizawa theorems, the tracking error $\mathbf{e_1(t)}$ tends to zero as time tends to infinite. Therefore, the proof of the Theorem 1 is completed.

For the second equation in Equation (8) $\dot{\mathbf{x}}_2(\mathbf{t}) = \mathbf{F} + \mathbf{\Sigma Gu(t)}$, the reference signal is $\mathbf{x_{2d}} = \mathbf{x_2} = \mathbf{B}^{-1}(-\mathbf{A} - \hat{\mathbf{d}} + \dot{\mathbf{x}}_{\mathbf{1d}} - \mathbf{K_1 e_1} - \mathbf{C_1 s_1} - \mathbf{E_1 sgn(s_1)})$. The tracking error is defined as follows:

$$\mathbf{e_2(t)} = \mathbf{x_2(t)} - \mathbf{x_{2d}(t)} \tag{27}$$

The sliding surfaces are defined in the form of:

$$\mathbf{s_2(t)} = \mathbf{e_2(t)} + \mathbf{K_2} \int_0^t \mathbf{e_2(t)d\tau} \tag{28}$$

Taking the derivative of $\mathbf{s_2(t)}$ with respect to time yields:

$$\mathbf{\dot{s}_2} = \mathbf{\dot{e}_2} + \mathbf{K_2 e_2} = \mathbf{\dot{x}_2} - \mathbf{\dot{x}_{2d}} + \mathbf{K_2 e_2} \tag{29}$$

Substituting Equation (8) into Equation (29) yields:

$$\mathbf{\dot{s}_2} = \mathbf{F} + \mathbf{\Sigma G u} - \mathbf{\dot{x}_{2d}} + \mathbf{K_2 e_2} \tag{30}$$

Assuming $\mathbf{\chi} = \mathbf{F} - \mathbf{\dot{x}_{2d}} + \mathbf{K_2 e_2} + \mathbf{C_2 s_2} + \mathbf{E_2 sgn(s_2)})$ and $\mathbf{Z} = (\mathbf{\Sigma G})^{-1}$, the estimation of $\mathbf{Z}$ is defined as $\mathbf{\hat{Z}}$. The estimation error is given by $\mathbf{\widetilde{Z}} = \mathbf{\hat{Z}} - \mathbf{Z}$. Taking the command input $\mathbf{u}$ in the form of:

$$\mathbf{u} = -\mathbf{\hat{Z}\chi} = -\mathbf{\hat{Z}}(\mathbf{F} - \mathbf{\dot{x}_{2d}} + \mathbf{K_2 e_2} + \mathbf{C_2 s_2} + \mathbf{E_2 sgn(s_2)}) \tag{31}$$

**Theorem 2.** *For the second equation in Equation (8), supposing the adaptive law is $\mathbf{\dot{\hat{Z}}} = \lambda \mathbf{s_2^T \chi sgn(G)}$, where $\lambda$ is a constant coefficient. If the command input is chosen in the form of Equation (31), the tracking error $\mathbf{e_2(t)}$ can asymptotically converge to zero, which means the faulty aircraft system is asymptotically stable.*

**Proof 2.** Selecting a Lyapunov candidate function as follows:

$$V_2 = \frac{1}{2}\mathbf{s_2^T s_2} + \frac{|\mathbf{\Sigma G}|}{2\lambda}\mathbf{\widetilde{Z}^T \widetilde{Z}} \tag{32}$$

. $\square$

The derivative of $V_2$ is:

$$\dot{V}_2 = \mathbf{s_2^T \dot{s}_2} + \frac{|\mathbf{\Sigma G}|}{\lambda}\mathbf{\widetilde{Z}^T \dot{\widetilde{Z}}} \tag{33}$$

Substituting Equations (29) and (31) into Equation (33) yields:

$$\begin{aligned}\dot{V}_2 &= \mathbf{s_2^T}(\mathbf{F} + \mathbf{\Sigma G u} - \mathbf{\dot{x}_{2d}} + \mathbf{K_2 e_2}) + \frac{|\mathbf{\Sigma G}|}{\lambda}\mathbf{\widetilde{Z}^T \dot{\hat{Z}}} \\ &= \mathbf{s_2^T}(\mathbf{\chi} - \mathbf{\Sigma G \hat{Z} \chi} - \mathbf{C_2 s_2} - \mathbf{E_2 sgn(s_2)}) + \frac{|\mathbf{\Sigma G}|}{\lambda}\mathbf{\widetilde{Z}^T \dot{\hat{Z}}}\end{aligned} \tag{34}$$

Substituting the adaptive law $\mathbf{\dot{\hat{Z}}} = \lambda \mathbf{s_2^T \chi sgn(G)}$ into Equation (34) yields:

$$\dot{V}_2 = \mathbf{s_2^T}(\mathbf{\chi} - \mathbf{\Sigma G \hat{Z} \chi} - \mathbf{c_2 s_2} - \mathbf{E_2 sgn(s_2)}) + |\mathbf{\Sigma G}|\mathbf{\widetilde{Z}^T s_2^T \chi sgn(G)} \tag{35}$$

Noting that $\mathbf{sgn(G)} = \mathbf{sgn(\Sigma G)}$. Therefore, Equation (35) can be rewritten as:

$$\begin{aligned}\dot{V}_2 &= \mathbf{s_2^T}(\mathbf{\chi} - \mathbf{\Sigma G \hat{Z} \chi} - \mathbf{C_2 s_2} - \mathbf{E_2 sgn(s_2)}) + \mathbf{s_2^T \Sigma G \widetilde{Z} \chi} \\ &= \mathbf{s_2^T}(\mathbf{\chi} - \mathbf{\Sigma G \hat{Z} \chi} + \mathbf{\Sigma G \widetilde{Z} \chi} - \mathbf{C_2 s_2} - \mathbf{E_2 sgn(s_2)}) \\ &= \mathbf{s_2^T}(\mathbf{\chi} - \mathbf{\Sigma G Z \chi} - \mathbf{C_2 s_2} - \mathbf{E_2 sgn(s_2)}) \\ &= -\mathbf{C_2 s_2}^2 - \mathbf{E_2}|\mathbf{s_2}| \leq 0\end{aligned} \tag{36}$$

According to the Lyapunov and LaSalle–Yoshizawa theorems, the tracking error $\mathbf{e_2(t)}$ tends to zero as time tends to infinite. Therefore, the proof of the Theorem 2 is completed. To sum up, the NASFTC controller $\mathbf{u}$ for the faulty aircraft system is designed as:

$$\begin{cases} \mathbf{u} = -\mathbf{\hat{Z}\chi} = -\mathbf{\hat{Z}}(\mathbf{F} - \mathbf{\dot{x}_{2d}} + \mathbf{K_2 e_2} + \mathbf{C_2 s_2} + \mathbf{E_2 sgn(s_2)}) \\ \mathbf{x_{2d}} = \mathbf{B}^{-1}(-\mathbf{A} - \mathbf{\hat{W}^T h} + \mathbf{\dot{x}_{1d}} - \mathbf{K_1 e_1} - \mathbf{C_1 s_1} - \mathbf{E_1 sgn(s_1)}) \end{cases} \tag{37}$$

where $\dot{\mathbf{W}}^{\mathbf{T}} = \mu \mathbf{s_1}\mathbf{h}$ and $\dot{\mathbf{Z}} = \lambda \mathbf{s_2}^{\mathbf{T}}\boldsymbol{\chi}\mathbf{sgn}(\mathbf{G})$ However, the switching term sgn($s$) is a discontinuous function. This switching term will cause chattering problem sometimes, which is unacceptable to the system. To eliminate the chattering problem, the saturation function $sat(s)$ is used to replace the discontinuous function sgn($s$). The saturation function is given by:

$$sat(s) = \begin{cases} 1 & s > \Delta \\ \dfrac{s}{\Delta} & |s| < \Delta \\ -1 & s < -\Delta \end{cases} \tag{38}$$

where $\Delta$ is a constant.

### 3.3. Approach Power Compensation System (APCS)

During the carrier landing process, the sideslip angle is expected to be zero to maintain a coordinated turn and the relationship between the three longitudinal angles, i.e., the flight path angle $\gamma$, AoA $\alpha$, and pitch angle $\theta$ can be described as:

$$\theta = \alpha + \gamma \tag{39}$$

With the guidance law mentioned in Section 2, the height error can be transformed into the flight path angle command. However, the input that is required by the longitudinal controller is the pitch angle command. From Equation (39), the changes between the pitch angle command and the flight path angle are synchronous as long as $\alpha$ is unchanging. The function of APCS is to maintain the AoA. Therefore, the height error can be transformed in the pitch angle command, which is required by the controller. The design steps of APCS are described as follows:

Assuming the tracking error of the AoA is:

$$e_\alpha = \alpha - \alpha_d \tag{40}$$

where $\alpha_d$ is the desired AoA. By introducing the information of vertical load $n_y$ and deflection of elevator $\delta_e$, the control scheme for APCS is designed by:

$$\delta_T = k_P e_\alpha + k_I \int e_\alpha + k_n n_y + k_{\delta_e} \delta_e \tag{41}$$

where $k_P$, $k_I$, $k_n$, and $k_{\delta_e}$ are constant coefficients.

## 4. Simulation Results and Discussion

In this section, a series of comparative simulations have been carried out to verify the effectiveness of the proposed NASFTC. The parameters of the carrier-based UAV's initial states are chosen as: $\mathbf{x_{10}} = \begin{bmatrix} \phi_0 & \theta_0 & \beta_0 \end{bmatrix} = \begin{bmatrix} 0° & -2.8° & 0° \end{bmatrix}$, $\mathbf{x_{20}} = \begin{bmatrix} p_0 & q_0 & r_0 \end{bmatrix} = \begin{bmatrix} 0 & 0 & 0 \end{bmatrix}$, $\mathbf{u_0} = \begin{bmatrix} \delta_{a0} & \delta_{e0} & \delta_{r0} \end{bmatrix} = \begin{bmatrix} 0 & 0 & 0 \end{bmatrix}$, $\alpha_0 = 8.1°$, $V = 70m/s$. The control parameters are as follows: $\mathbf{K_1} = diag(20, 15, 20)$ and $\mathbf{K_2} = diag(15, 10, 15)$ for sliding surfaces; $\mathbf{C_1} = diag(5, 20, 5)$, $\mathbf{E_1} = diag(1, 2, 1)$, $\mathbf{C_2} = diag(3, 12, 3)$, and $\mathbf{E_2} = diag(1, 2, 1)$ for NASFTC; $\mu = 0.5$ and $\lambda = 2$ for adaptive laws; $k_P = 20$, $k_I = 5$, $k_n = 3$, and $k_{\delta_e} = 0.1$ for APCS.

To better demonstrate the superiority of the proposed controller, a PID controller is introduced for comparison. Simulations of three flight conditions are conducted. The three flight conditions include "normal states", which indicates the carrier landing of the undamaged aircraft with PID controller; "Faulty aircraft with NASFTC", which indicates the carrier landing of faulty aircraft with NASFTC; and "Faulty aircraft with PID", which indicates the carrier landing of faulty aircraft with PID controller.

The following simulation results are divided into three scenarios. Scenario 1 considers only the actuator partial loss fault. Scenario 2 considers only the additional unknown fault. Scenario 3 considers both the actuator partial loss fault and the additional unknown fault.

### 4.1. Scenario 1: Only Actuator Partial Loss Fault Is Considered

In this part, the comparative simulation results of only actuator partial loss fault are presented. When $t > 1\ s$, the aircraft undergoes actuator partial loss faults, the actuator effectiveness matrix is given by:

$$\Sigma(\mathbf{t}) = diag(0.5, 0.5, 0.5) \tag{42}$$

Equation (42) denotes that all the actuators lose half of their effectiveness 1 s after the starting of the landing process. However, the happening time and value of the actuator faults are unknown to the UAV system. The results of the two-dimensional trajectory tracking error are shown in Figure 5. It can be seen that in normal states, the PID controller can command the carrier-based UAV to track the reference trajectory. When considering the actuator partial loss fault, the control performance of the PID controller deteriorates seriously, which means the actuator faults will greatly weaken the aircraft's tracking ability. It can also be noticed that compared to lateral direction, the longitudinal tracking ability declines more obviously. The reason is that the longitudinal tracking is more difficult and has higher requirements for actuators. However, the proposed NASFTC method can effectively solve the problem that is induced by the actuator faults. The trajectory tracking error is effectively reduced and is confined to a small enough interval before touchdown.

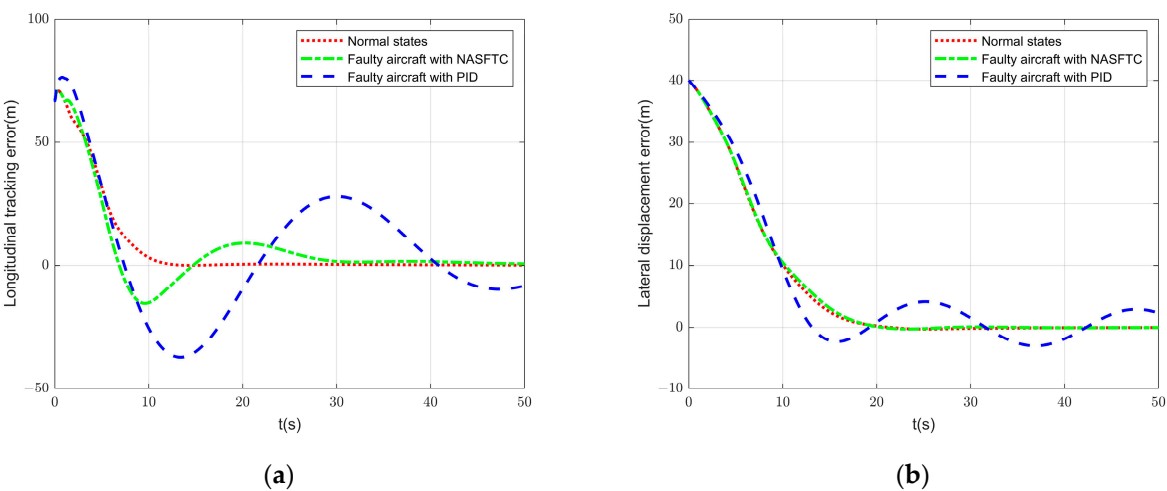

**Figure 5.** Two-dimensional trajectory tracking errors: (**a**) longitudinal tracking error; and (**b**) lateral tracking error.

The credit for this improvement is the adaptive laws possessed by NASFTC. The adaptive laws can automatically change the control parameter according to the external changes, thus compensating for the errors that are induced by faults. The reason can be further concluded from the attitude angles and outputs of actuators, which are shown in Figures 6 and 7, respectively.

The variations of attitude angles are presented in Figure 6. From Figure 6a,b, it can be seen that the AoA is almost constant, and the sideslip angle remains at zero, which exactly meets the requirements of the carrier landing. Figure 6c,d demonstrates that even though there actuator faults exist, the NASFTC method still can stabilize the UAV's attitude angles in time, while the attitude angles of PID method fluctuates obviously.

The outputs of actuators are presented in Figure 7. It can be seen that the actuator loses half of its effectiveness after 1 s. Due to the actuator faults, the PID method is unable to give a proper control signal. However, the adaptive laws in NASFTC can make the actuators operate more effectively to compensate for the partial loss. This compensation is especially apparent in Figure 7b since the longitudinal tracking has higher requirements for actuators.

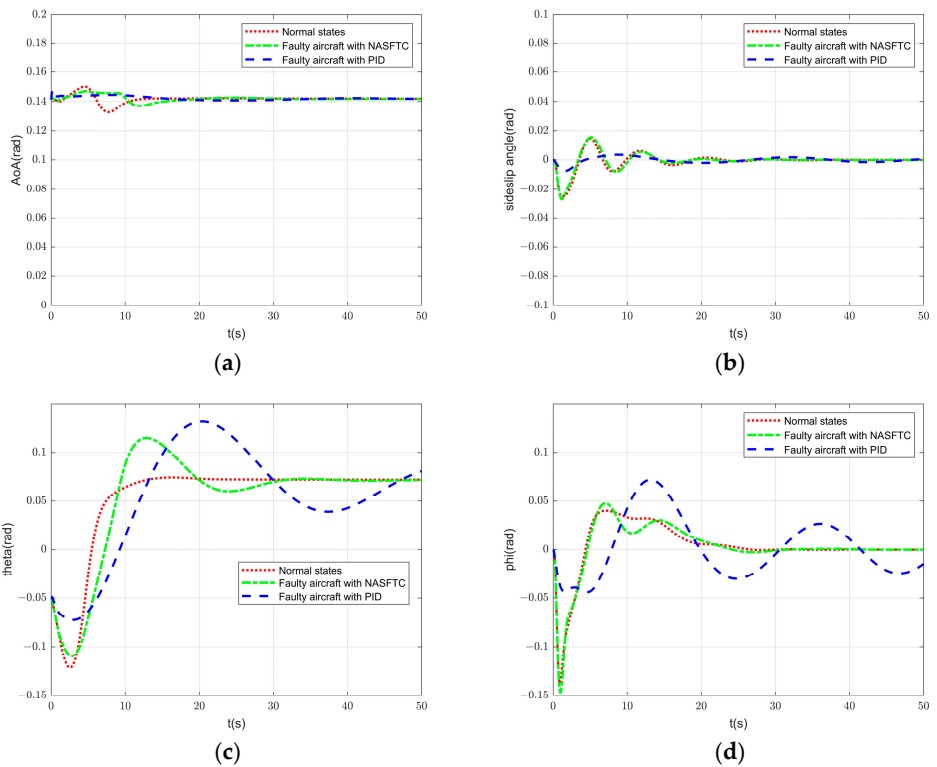

**Figure 6.** Aircraft attitude angles. (**a**) AoA; (**b**) sideslip angle; (**c**) pitch angle; and (**d**) roll angle.

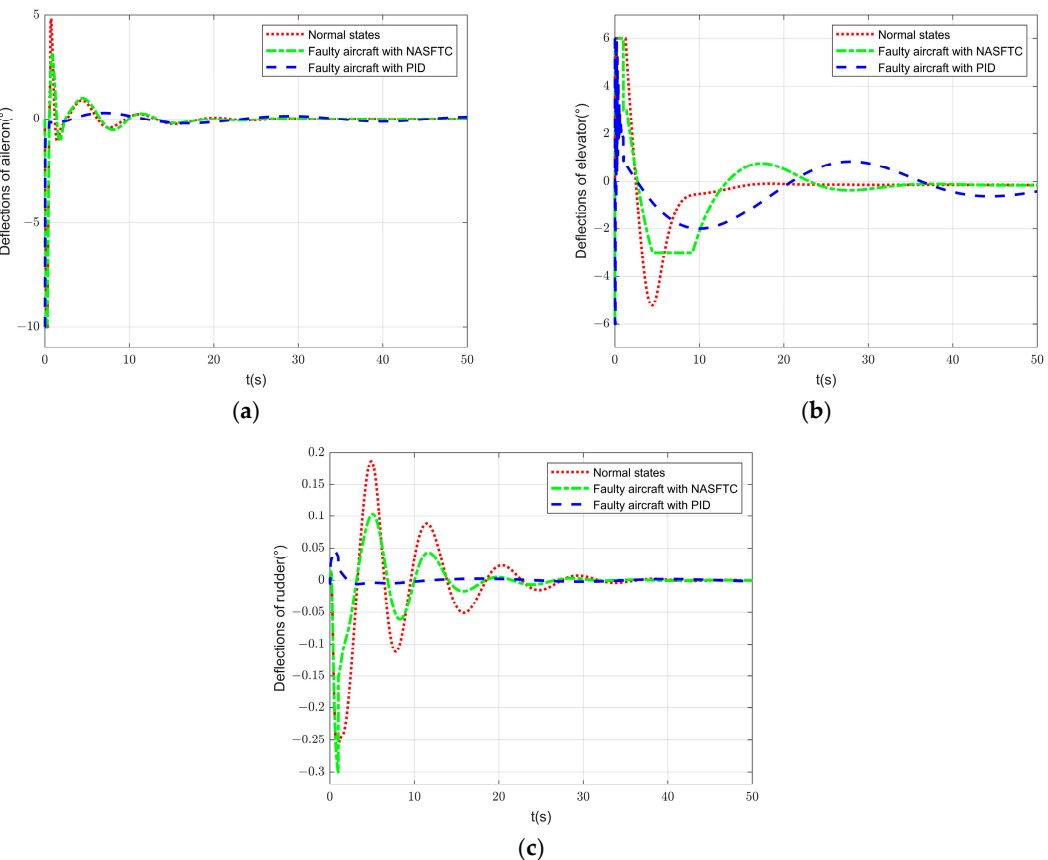

**Figure 7.** Deflections of actuators. (**a**) deflections of aileron; (**b**) deflections of elevator; (**c**) deflections of rudder.

*4.2. Scenario 2: Only Additional Unknown Fault Is Considered*

In this part, the comparative simulation results of only additional unknown faults are presented. When the carrier landing process begins (*t* > 0 s), the carrier-based UAV undergoes the additional unknown faults. The detailed expressions of the additional unknown faults are described by the following equations. However, these detailed expressions are unknown to the UAV flight system during the simulation process.

$$\mathbf{d(t, x_1, x_2)} = \begin{bmatrix} d_1 & d_2 & d_3 \end{bmatrix}^T \tag{43}$$

$$\begin{cases} d_1 = 0.25(p + \phi) + 0.065 \sin t \\ d_2 = 4.5(q + \theta) + 0.2 \sin t \\ d_3 = 2.5(r + \psi) + 0.1 \sin t \end{cases} \tag{44}$$

The two-dimensional trajectory tracking results are demonstrated in Figure 8. Compared to the actuator faults, the additional unknown faults have a larger impact on the UAV, especially in the longitudinal direction.

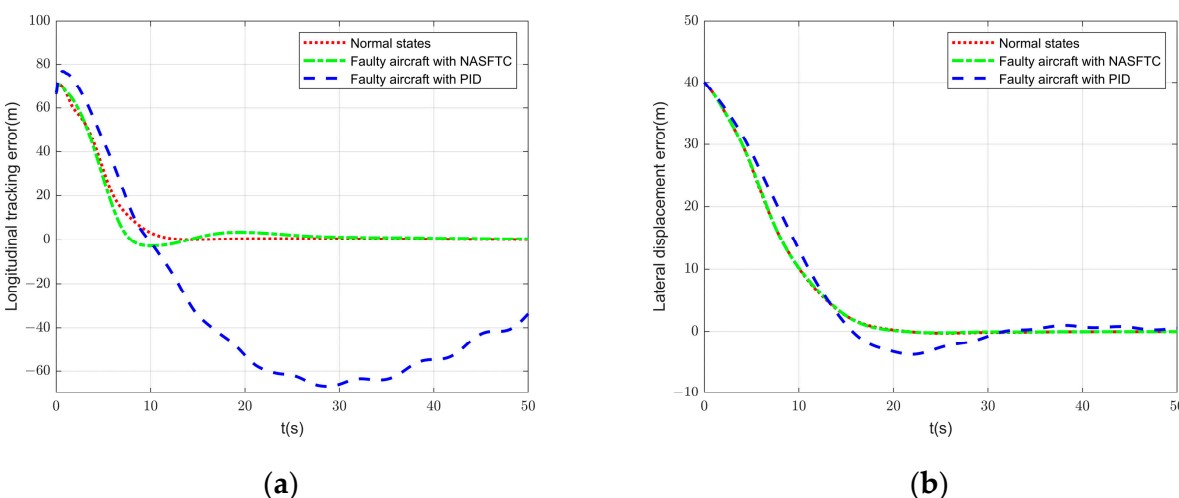

(**a**)　　　　　　　　　　　　　　　　　　　　(**b**)

**Figure 8.** Two-dimensional trajectory tracking errors: (**a**) longitudinal tracking error; and (**b**) lateral tracking error.

However, when the proposed NASFTC method is utilized, the additional unknown faults are compensated and the aircraft can achieve a safe carrier landing. The reason is that the RBF neural network has successfully estimated the additional unknown faults and made proper compensations, which is illustrated in Figure 9.

The compensation function of the RBF neural network can also be concluded from the attitude angles and the outputs of actuators, which are shown in Figures 10 and 11, respectively.

Figure 10 shows the variations of the attitude angles. Figure 10a,b shows that all the methods can keep the AoA and sideslip angle nearly at constants as required. However, without appropriate control methods, the additional unknown faults will cause large fluctuations in the pitch angle and roll angle, which is shown in Figure 10c,d.

The outputs of actuators are presented in Figure 11. It shows that the additional unknown faults will cause continuous fluctuations of the actuators. These fluctuations are extremely dangerous and are the immediate cause of the landing failure. After the RBF neural network is introduced, the impact of additional unknown faults is largely eliminated. Despite some minor differences, the outputs of actuators are similar to those of the normal states.

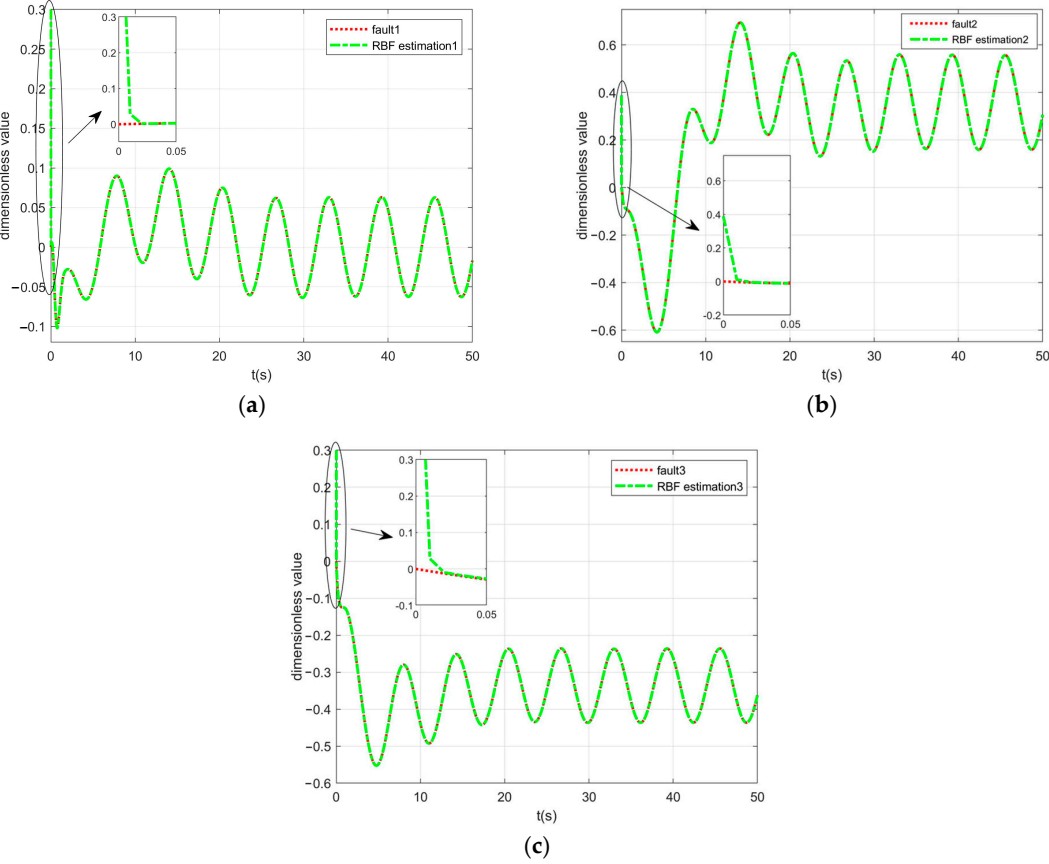

**Figure 9.** Additional unknown faults and RBF estimations. (**a**) Fault 1 and estimation; (**b**) Fault 1 and estimation; and (**c**) Fault 1 and estimation.

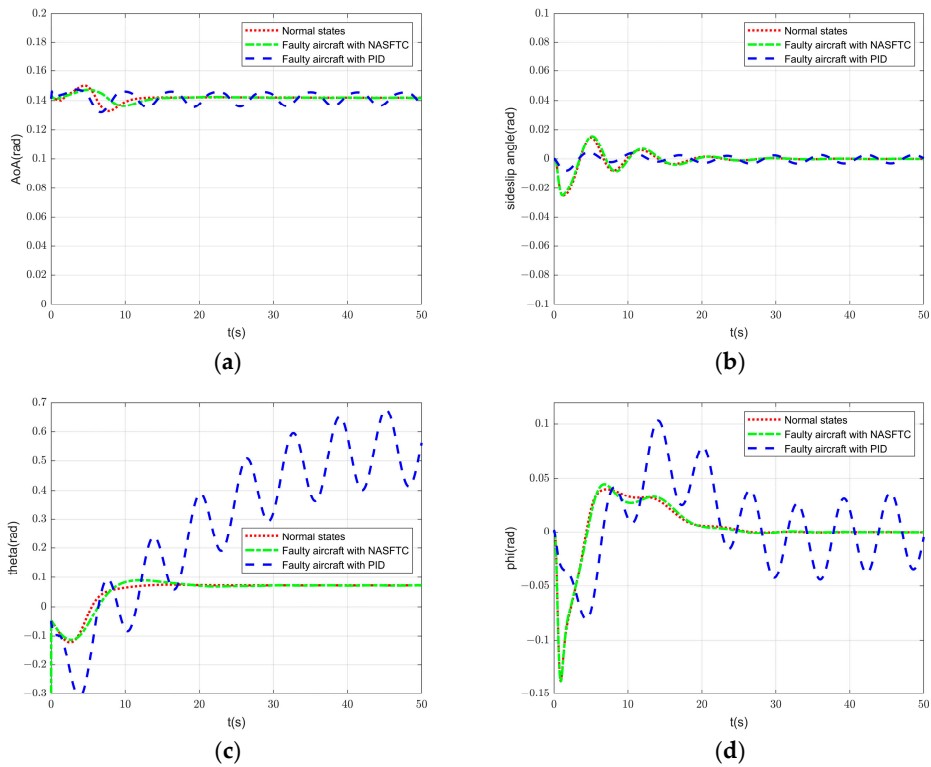

**Figure 10.** Aircraft attitude angles: (**a**) AoA; (**b**) sideslip angle; (**c**) pitch angle; and (**d**) roll angle.

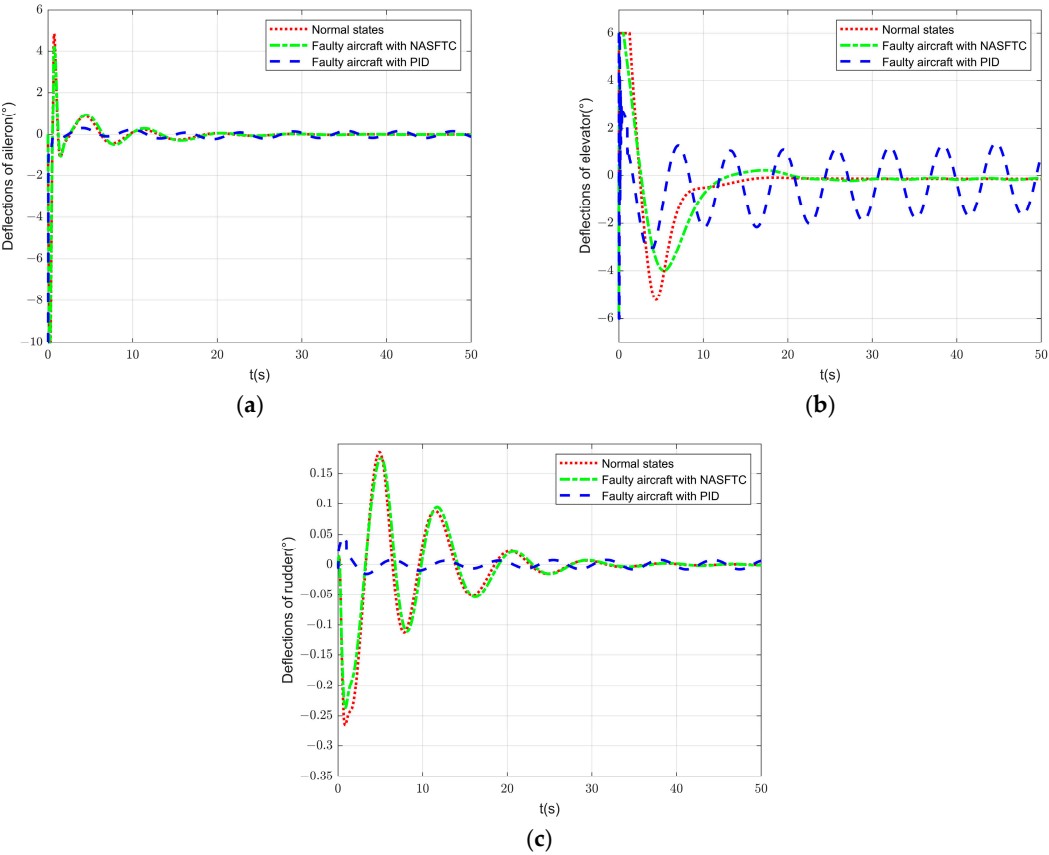

**Figure 11.** Deflections of actuators: (**a**) deflections of aileron; (**b**) deflections of elevator; and (**c**) deflections of rudder.

### 4.3. Scenario 3: Both Actuator Partial Loss and Additional Unknown Faults Are Considered

In this part, both the actuator partial loss fault and the additional unknown fault are injected into the UAV system. The fault structures and injection times are the same with Scenario 1 and 2. Likewise, both of the faults are unknown to the system. The simulation results of this case are presented in Figures 12–14. Figure 12 demonstrates the two-dimensional trajectory tracking error, from which we can see that the tracking performance of the PID controller is far beyond satisfaction. It can also be found that the impact of the additional unknown fault plays a dominant role.

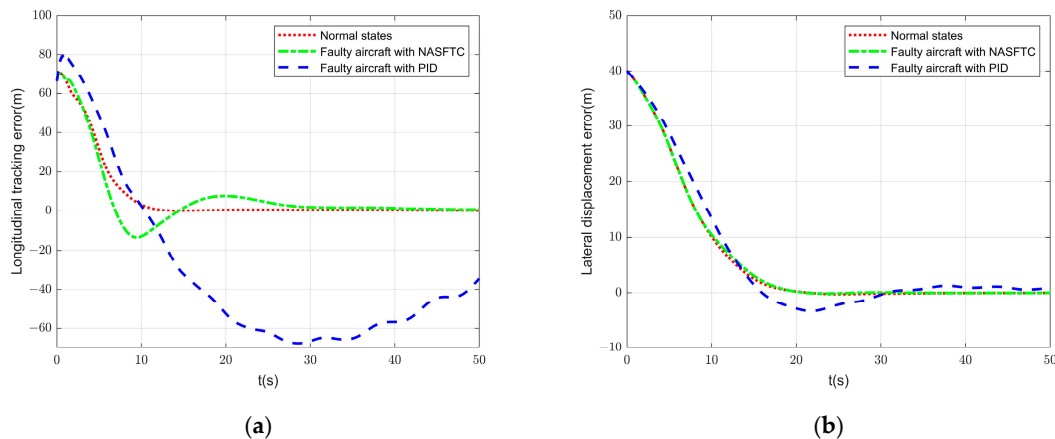

**Figure 12.** Two-dimensional trajectory tracking errors: (**a**) longitudinal tracking error; and (**b**) lateral tracking error.

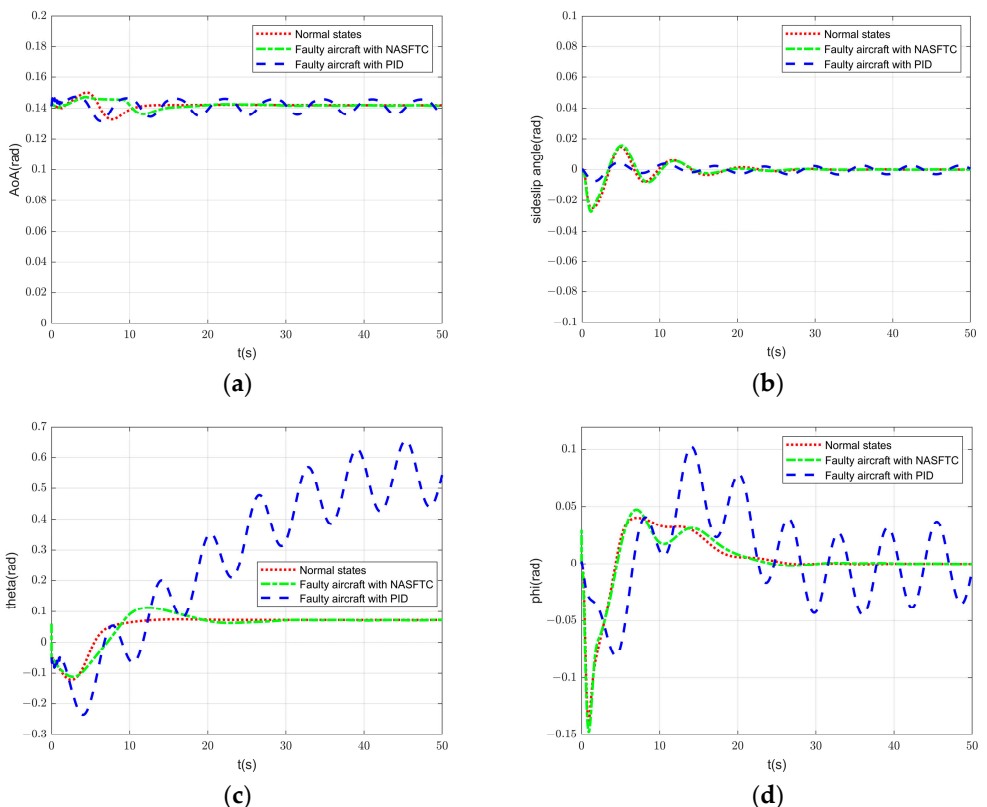

**Figure 13.** Aircraft attitude angles: (**a**) AoA; (**b**) sideslip angle; (**c**) pitch angle; and (**d**) roll angle.

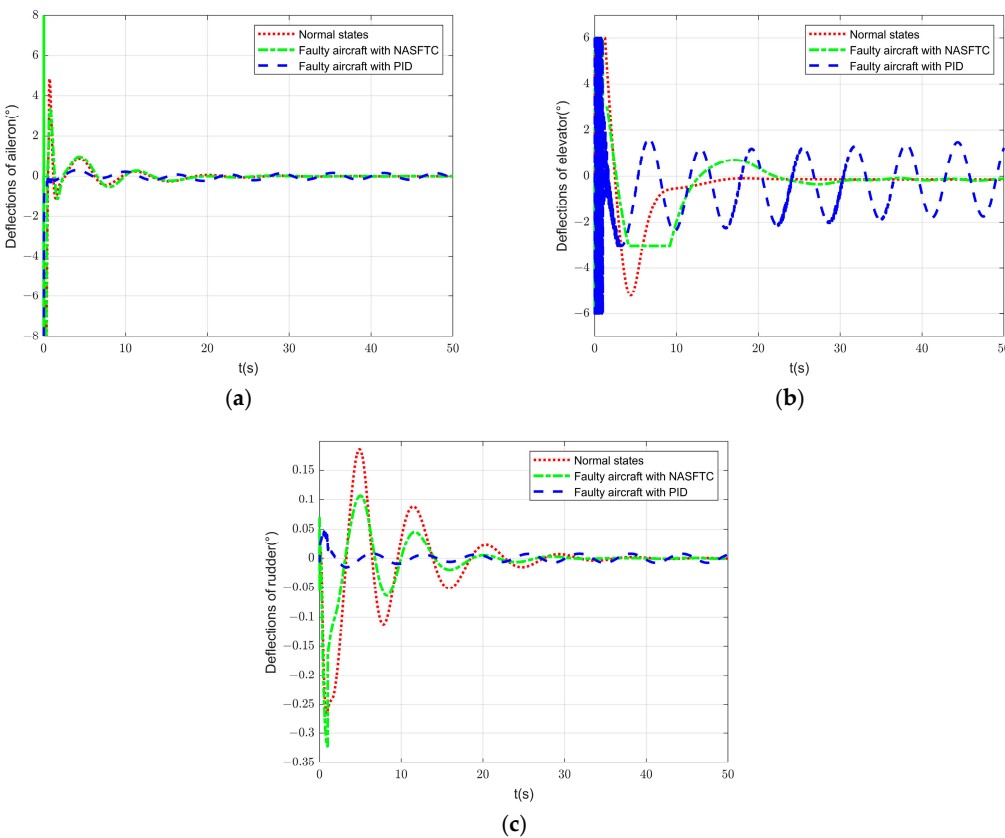

**Figure 14.** Deflections of actuators: (**a**) deflections of aileron; (**b**) deflections of elevator; and (**c**) deflections of rudder.

The quantitative error criteria of tracking errors are presented in Table 1 to better demonstrate the controller performance. The integrated absolute error (IAE) and integrated time absolute error (ITAE) are calculated for comparisons. It can be seen that the values IAE and ITAE of NASFTC are much less than that of the PID in both longitudinal and lateral channels.

**Table 1.** Quantitative comparisons of tracking errors.

| Channel | Index | Normal States | NASFTC | PID |
|---|---|---|---|---|
| Longitudinal channel | IAE | 358 | 460 | 2456 |
| | ITAE | 1454 | 3952 | 64715 |
| Lateral channel | IAE | 294 | 297 | 359 |
| | ITAE | 1543 | 1558 | 2890 |

This is because the additional unknown faults are persistent fluctuations related to time, which puts a huge burden on the controller. With the superposition influence of these two faults, the carrier-based UAV has been completely unable to track the gliding trajectory, and in the condition "Faulty aircraft with PID", there even occurs the chattering problem in the elevator, which is shown in Figure 14b. Therefore, without a better controller, the carrier landing of the aircraft will be extremely dangerous in the presence of these two faults.

The NASFTC method that is proposed in this paper can fix the problem caused by the multiple faults. From Figure 12, we can see that in the longitudinal direction, there are only slight fluctuations in the tracking trajectory. The result of the lateral tracking trajectory is more ideal, almost consistent with the normal states. The reason for this improvement can be concluded from Figures 13 and 14. With the adaptive laws compensating for the actuator fault and the RBF neural network compensating for the additional unknown fault, the change of attitude angles and actuator outputs of the aircraft are approaching the normal states.

## 5. Conclusions

Due to the harsh external environment, faults will occur in the carrier-based UAV during the landing process, which can directly lead to a landing failure. Motivated by the practical necessity of high accuracy and security for automatic carrier landing with the presence of multiple faults, this paper establishes an automatic carrier landing system (ACLS) with the neural network-based adaptive sliding mode fault-tolerant control (NAS-FTC) method. In the system, the longitudinal and lateral guidance laws are developed with Hdot guidance law and PID method, respectively; the faulty aircraft model is established, which includes the aircraft model, the actuator partial loss fault model and the additional unknown fault model; the approach power compensation system (APCS) is designed to maintain the AoA; and the controllers have been designed in both longitudinal and lateral channels for trajectory tracking and faults compensation. The controller consists of three parts, namely the RBF neural network, adaptive law, and sliding mode method, each with its function. The function of adaptive laws is to deal with the actuator partial loss fault, which is achieved by automatically updating the controller parameters according to the fault. The RBF neural network is used to deal with the additional unknown fault, which is achieved by predicting and compensating the floating error. The sliding mode method can ensure the overall tracking of the desired glide slope trajectory. The stability analysis of the system is carried out by the Lyapunov function.

To verify the efficiency of the proposed NASFTC method, comparative simulations of three different scenarios are conducted. The three scenarios include undamaged aircraft with PID controller, faulty aircraft with PID controller, and faulty aircraft with NASFTC. Comparative results demonstrate that the proposed NASFTC method can effectively eliminate the effect that is induced by actuator partial loss fault and additional unknown faults, thus accurately commanding the carrier-based UAV to land on the deck.

**Author Contributions:** Conceptualization, D.L., Z.Y. and Z.K.; methodology, Z.Y.; software, Z.Y. and Z.K.; validation, C.Z., Z.Y. and Z.K.; formal analysis, Z.Y. and Z.K.; investigation, Z.Y. and H.S.; resources, Z.K.; data curation, C.Z., Z.Y. and Z.K.; writing—original draft preparation, Z.Y.; writing—review and editing, Z.Y., Z.K. and H.S.; visualization, Z.Y. and Z.K.; supervision, D.L.; project administration, D.L.; funding acquisition, D.L. and C.Z. All authors have read and agreed to the published version of the manuscript.

**Funding:** This research was funded by the National Natural Science Foundation of China (Grant Number 11972059) and National Key Research and Development Project (Grant Number 2020YFC1512500).

**Data Availability Statement:** Not applicable.

**Conflicts of Interest:** The authors declare no conflict of interest.

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
