# Peer review of "Fault-Tolerant Control for Carrier-Based UAV Based on Sliding Mode Method"

_drones, doi:10.3390/drones7030194_

Round 1

Reviewer 1 Report

This paper establishes the dynamic model, actuator failure model, additional unknown fault model and control framework of aircraft carrier automatic landing system (ACLS) of carrier-based UAV. A large number of experiments have been carried out to prove the superiority of the proposed method. I think it's an interesting piece of work. But there are still certain problems with this manuscript that need to be revised:

1. I noticed that the author used only numbers in some citations of the article, and lacked [ ] 

2. Why is Equation 2 and Equation 3 the same

Reviewer 2 Report

- How you have validated the dynamic model of the UAV?

- Please explicitly mention the novelty of the proposed FTC scheme.

- Mention the results quantitatively. e.g. 'strong fault-tolerant ability' (How much strong is strong?) 'great effectiveness' (How much great is great)?

- Please arrange the references in the ascending order. So, the first reference should be [1], [2] and so on. Also, enclose the reference number in [].

- Most of the equations have been written in bold. Please follow standard notations e.g. matrices are denoted by bold face upper case letters.

- In Section 3.2 (Controller design), briefly mention the Sliding mode controller with reference to 10.3390/electronics12040876  

- How the actuator effectiveness matrix given in (38) has been calculated. Similarly, how the additional unknown faults in (40) are computed?

- In Section 1 (Introduction), mention various sources of faults; in sensors and actuators, in plant conditions, bad tuning of controller parameters, process abnormalities, damage in equipment and environmental changes with reference to DOI: 10.1371/journal.pone.0256491.

- Literature review needs to be updated. Include recent references on sensor and actuator fault tolerant control e.g. 10.1007/s42835-019-00277-9 (2019)

- There are many interesting results (Figures 5-14), however the discussion needs to be elaborated and should be more critical and conclusive.

- Please use error criteria for analysing the tracking errors: ITAE, IAE, ISE, ITSE

Reviewer 3 Report

Find my comments in the file attached.

Round 2

Reviewer 2 Report

The authors have addressed all of my comments made earlier. The paper may be accepted for publication.

Reviewer 3 Report

Great job!